# Ceftazidime–Avibactam Improves Outcomes in High-Risk Neutropenic Patients with *Klebsiella pneumoniae* Carbapenemase-Producing Enterobacterales Bacteremia

**DOI:** 10.3390/microorganisms12010195

**Published:** 2024-01-18

**Authors:** Fabián Herrera, Diego Torres, Ana Laborde, Rosana Jordán, Noelia Mañez, Lorena Berruezo, Sandra Lambert, Nadia Suchowiercha, Patricia Costantini, Andrea Nenna, María Laura Pereyra, José Benso, María Luz González Ibañez, María José Eusebio, Laura Barcán, Nadia Baldoni, Lucas Tula, Inés Roccia Rossi, Martín Luck, Vanesa Soto, Verónica Fernández, Alberto Ángel Carena

**Affiliations:** 1Centro de Educación Médica e Investigaciones Clínicas (CEMIC), Buenos Aires C1431, Argentina; diegots23@hotmail.com (D.T.); alberto.carena390@gmail.com (A.Á.C.); 2Fundación para Combatir la Leucemia, Buenos Aires C1114, Argentina; anarlaborde@gmail.com (A.L.); lgonzalez@fundaleu.org.ar (M.L.G.I.); 3Infectious Diseases Service, Hospital Británico de Buenos Aires, Buenos Aires C1280, Argentina; rosanajordan61@gmail.com (R.J.); joule_cba@hotmail.com (M.J.E.); 4Infectious Diseases Section, Internal Medicine Department, Hospital Italiano de Buenos Aires, Buenos Aires C1199, Argentina; noelia.manez@hospitalitaliano.org.ar (N.M.); laura.barcan@hospitalitaliano.org.ar (L.B.); 5Infectious Diseases Service, Hospital HIGA Rodolfo Rossi, La Plata B1902, Argentina; lvberruezo@gmail.com (L.B.); nadiabaldoni@hotmail.com (N.B.); 6Infectious Diseases Service, Hospital El Cruce, Buenos Aires B1888, Argentina; sandralambert2000@yahoo.com.ar (S.L.); lft1969@yahoo.com.ar (L.T.); 7Infectious Diseases Service, Hospital HIGA Gral. San Martín, La Plata B1900, Argentina; n.suchowiercha@gmail.com (N.S.); ainesines@hotmail.com (I.R.R.); 8Infectious Diseases Service, Instituto de Oncología Angel H. Roffo, Buenos Aires C1417, Argentina; patricostantini@gmail.com (P.C.); martinlu100@hotmail.com (M.L.); 9Infectious Diseases Service, Hospital Municipal de Oncología Marie Curie, Buenos Aires C1405, Argentina; andrea_nenna@yahoo.com.ar (A.N.); vanesitasoto@gmail.com (V.S.); 10Infectious Diseases Service, Hospital Universitario Austral, Buenos Aires B1629, Argentina; laurapereyra@yahoo.com.ar; 11Infectious Diseases Section, Internal Medicine Department, Hospital Italiano de San Justo, Buenos Aires C1198, Argentina; jose.benso@hospitalitaliano.org.ar (J.B.); veronica.fernandez@hospitalitaliano.org.ar (V.F.)

**Keywords:** ceftazidime–avibactam, KPC-producing, bacteremia, neutropenia

## Abstract

Few studies have evaluated the efficacy of ceftazidime–avibactam (CA) for *Klebsiella pneumoniae* carbapenemase-producing Enterobacterales bacteremia (KPC-PEB) in high-risk neutropenic patients. This is a prospective multicenter observational study in high-risk neutropenic patients with multi-drug resistant Enterobacterales bacteremia. They were compared according to the resistance mechanism and definitive treatment provided: KPC-CPE treated with CA (G1), KPC-CPE treated with other antibiotics (G2), and patients with ESBL-producing Enterobacterales bacteremia who received appropriate definitive therapy (G3). Thirty-day mortality was evaluated using a logistic regression model, and survival was analyzed with Kaplan–Meier curves. A total of 238 patients were included: 18 (G1), 52 (G2), and 168 (G3). *Klebsiella* spp. (60.9%) and *Escherichia coli* (26.4%) were the Enterobacterales most frequently isolated, and 71% of the bacteremias had a clinical source. The resistance profile between G1 and G2 was colistin 35.3% vs. 36.5%, amikacin 16.7% vs. 40.4%, and tigeclycline 11.1% vs. 19.2%. The antibiotics prescribed in combination with G2 were carbapenems, colistin, amikacin, fosfomycin, tigecycline, and fluoroquinolones. Seven-day clinical response in G1 vs. G2 vs. G3 was 94.4% vs. 42.3% vs. 82.7%, respectively (*p* < 0.001). Thirty-day overall mortality in G1 vs. G2 vs. G3 was 22.2% vs. 53.8% vs. 11.9%, respectively (*p* < 0.001), and infection-related mortality was 5.5% vs. 51.9% vs. 7.7% (*p* < 0.001). The independent risk factors for mortality were Pitt score > 4: OR 3.63, 95% CI, 1.18–11.14 (*p* = 0.025) and KPC-PEB treated with other antibiotics: OR 8.85, 95% CI, 2.58–30.33 (*p* = 0.001), while 7-day clinical response was a protective factor for survival: OR 0.02, 95% CI, 0.01–0.08 (*p* < 0.001). High-risk neutropenic patients with KPC-CPE treated with CA had an outcome similar to those treated for ESBL-producing Enterobacterales, with higher 7-day clinical response and lower overall and infection-related mortality than those treated with other antibiotics. In view of these data, CA may be considered the preferred therapeutic option for KPC-PEB in high-risk neutropenic patients.

## 1. Introduction

Enterobacterales are the leading cause of bacteremia in high-risk neutropenic patients [1,2,3,4]. Likewise, growing antibiotic resistance is a major concern worldwide [4]. In this regard, the prevalence of carbapenem-resistant Enterobacterales, mainly mediated by *Klebsiella pneumoniae* carbapenemase (KPC) production, has become a challenging problem in these patients. Several studies showed that the prevalence varies with the region or country. A prospective multicenter cohort study on patients with hematological malignancies (Hematologic Malignancies Associated Bloodstream Infections Surveillance [HeMABIS] registry–Sorveglianza Epidemiologica InFezioni nelle EMopatie [SEIFEM] group) was performed in Italy between 2016 and 2018 [5]. They compared the epidemiology of bacteremia caused by Gram-negative bacteria (GNB) and their antimicrobial resistance profiles with a previous study conducted between 2009 and 2012. A significant increase in carbapenem-resistant isolates was observed (7.81% vs. 13.95%, *p* < 0.05). On the other hand, a prospective multicenter study conducted in Argentina between 2014 and 2020 (ROCAS study), which included 1277 bacteremias in onco-hematological patients, described the etiology and resistance patterns among GNB. Resistance to meropenem, mainly mediated by KPC-producing Enterobacterales, in patients with hematological malignancies and hematopoietic stem cell transplant patients was 18.4% vs. 26.4% (*p* = 0.016), respectively. In addition, resistance to colistin, amikacin, and tigecycline was 6.5% vs. 9%, 9.3% vs. 15.4%, and 7.2% vs. 16.9% [6].

During the COVID-19 pandemic, a significant increase in metallo-beta-lactamases (MBL) was observed. In a cohort of 1364 patients with hematological malignancies or hematopoietic stem cell transplants, a substantial increase was seen in bacteremia caused by GNB and multidrug-resistant GNB (MDR-GNB), reaching 69.9% and 40.4%, respectively, especially in the second and third waves compared to the pre-pandemic period. Likewise, an increase in extended-spectrum beta-lactamases (ESBL) and MBL-producing Enterobacterales was reported up to 24% and 3.7%, respectively [7].

These infections have a significant clinical impact, since, in patients with carbapenemase-producing Enterobacterales bacteremia, the 30-day mortality rate can be extremely high compared to patients with Enterobacterales bacteremia with no resistance mechanisms or even with ESBL producers [8]. A large study performed in Argentina on 532 Enterobacterales bacteremia episodes found that 30-day mortality among patients with Enterobacterales susceptible or resistant to third-generation cephalosporins vs. carbapenem-resistant was 17.6% vs. 17.9%, vs. 54.1% (*p* < 0.001) [9].

In view of this scenario, the choice of empirical antibiotic therapy in high-risk neutropenic patients is challenging in clinical practice, since it has been widely demonstrated that inadequate empirical antibiotic therapy significantly increases mortality in this population [9,10,11,12].

In recent years, new β-lactam–β-lactamase inhibitor combinations have become available for the treatment of KPC-producing Enterobacterales, including ceftazidime–avibactam, meropenem–vaborbactam, and imipenem–relebactam. However, since no randomized controlled trials have evaluated their clinical efficacy for these infections, treatment evidence is based on in vitro data, in vivo animal models, and observational cohort studies.

The IDSA and ESCCMID guidelines recommend one of these antibiotics for the treatment of severe carbapenem-resistant Enterobacterales infections if active in vitro, especially for KPC-producing Enterobacterales [13,14]. In addition, they recommend ceftazidime–avibactam as the preferred treatment for OXA-48-type-carbapenemase-producing Enterobacterales, since this is the only drug that demonstrated adequate in vitro activity. Despite the lack of an empirical approach, they suggest that it should be based on variables such as the pathogens most likely involved, severity and source of infection, and severe immune compromise, among others [14]. Several real-life studies have been conducted with ceftazidime–avibactam for KPC-producing Enterobacterales infections and OXA-48-type. They have all demonstrated superiority over polymyxin-based regimens or other antibiotics. Many included patients with hematological malignancies, hematopoietic stem cell transplants, and neutropenia, but they failed to analyze them separately from the total cohort [15,16,17,18,19]. Few retrospective studies have been published on patients with cancer, hematological malignancies, and hematopoietic stem cell transplants, and have shown the superiority of ceftazidime–avibactam over other antibiotics [20,21,22,23]. To our knowledge, however, no prospective multicenter studies have evaluated the clinical impact of ceftazidime–avibactam vs. other antibiotics on these infections exclusively in high-risk neutropenic patients.

The aim of this study was to assess the outcome and predictor factors for mortality of high-risk neutropenic patients with KPC-producing Enterobacterales bacteremia who received definitive therapy with ceftazidime–avibactam vs. other antibiotics. We further aimed to compare them with patients who received definitive therapy for ESBL-producing Enterobacterales bacteremia, since mortality may be similar to that of patients with no resistance mechanisms. Thus, the clinical impact of ceftazidime–avibactam on this population can be widely analyzed.

## 2. Materials and Methods

### 2.1. Setting, Patients, and Study Design

A multicenter prospective study was performed in 11 referral teaching centers (6 private and 5 public) specialized in the care of oncological and hematopoietic stem cell transplant patients in Argentina.

All episodes of initial Enterobacterales bacteremia (defined as the first episode of bacteremia experienced during an admission) in adult high-risk neutropenic patients (≥18 years of age) with hematological malignancies or hematopoietic stem cell transplant, who were managed as inpatients from May 2014 to April 2023, were evaluated. The following inclusion criteria were met: (a) the patient presented with ESBL-producing Enterobacterales or KPC-producing Enterobacterales monomicrobial bacteremia; (b) the patient had received at least 48 h of appropriate definitive antibiotic therapy. Patients with polymicrobial bacteremia, and those receiving palliative care or having a clinical source that required surgical procedures, were excluded from the analysis.

The study patients were divided into three groups: those with KPC-producing Enterobacterales bacteremia treated with ceftazidime–avibactam (G1), those with KPC-producing Enterobacterales bacteremia treated with other antibiotics (G2), and those treated for ESBL-Enterobacterales bacteremia (G3). Patients were identified by the Infectious Diseases Services in each center, which evaluate all individuals hospitalized with hematological malignancies and hematopoietic stem cell transplants who develop febrile neutropenia. 

Patients were included in the study at the time of positive blood culture, whether they had started empirical antibiotic therapy or not, and were then prospectively followed on a daily basis by direct patient care. Data were obtained from electronic and paper medical records and direct patient care, with a double check made with microbiological records from the laboratory. Variables included patients’ characteristics, type of hematological malignancy and hematopoietic stem cell transplant, stage of underlying disease, neutropenia duration, immunosuppressant drugs, previous and recent colonization with KPC-producing Enterobacterales, type of prior antibiotic use, Enterobacterales isolates with their resistance mechanisms and resistance profile, clinical source of bacteremia, Pitt and APACHE-II scores, type of antibiotic prescribed as monotherapy and combined, 7-day clinical response, 7- and 30-day mortality, infection-related mortality, and length of hospitalization prior to bacteremia. Empirical individualized antibiotic therapy was started based on the patient’s clinical and epidemiological features, according to the institutional guidelines of each center. The investigator in charge of the study chose definitive therapy based on the Enterobacterales isolates and their antibiotic resistance profile. Patients were followed for 30 days after the episode (by direct patient care in hospitalized cases, or by phone calls in the case of discharged patients), or until the patient’s death, provided that it happened before (assessed by direct patient care in patients still hospitalized or by a local healthcare database in each center).

### 2.2. Definitions

Neutropenia was defined as an absolute neutrophil count < 500 cells/mm^3^. High-risk febrile neutropenia was defined according to the Multinational Association for Supportive Care in Cancer (MASCC) score < 21 and one or more clinical criteria [24,25]. The clinical source of bacteremia was determined based on the isolation of Enterobacterales from the suspected source and/or the associated clinical signs and symptoms. It was classified according to the US CDC criteria [26].

High doses of corticosteroids were defined as prednisone (or equivalent) in doses ≥ 20 mg/day for a period ≥ 2 weeks prior to bacteremia, and the use of biological agents and/or anti-lymphocyte therapies, with these drugs administered within 6 months prior to bacteremia. KPC-producing Enterobacterales colonization was defined as “previous” when it occurred within six months prior to hospitalization and “recent” when it was detected within the week prior to the episode of bacteremia.

Bacteremia was classified as nosocomial, healthcare-associated, or community-acquired according to Friedman et al. [27]. Breakthrough bacteremia was defined as an episode of continuous or new-onset bacteremia in a patient receiving appropriate antibiotics for the microorganism recovered from blood cultures.

Septic shock was defined as the need for vasopressors to maintain mean arterial pressure ≥ 65 mmHg and serum lactate level > 18 mg/dL [28]. Infection severity and mortality probability were defined using Pitt and APACHE-II scores. Empirical antibiotic therapy was considered appropriate provided that it was started immediately after blood cultures were drawn and one or more antibiotics used were active in vitro against the isolated bacteria, with adequate dosing and dose interval. In patients with ESBL-producing Enterobacterales, empirical or definitive antibiotic therapy with piperacillin/tazobactam or cefepime monotherapy was considered inappropriate [29]. In patients with isolation of any Enterobacterales species, empirical or definitive therapy with tigecycline as monotherapy was deemed inappropriate. Clinical response on day 7 of antibiotic therapy was defined as absence of fever for at least 4 days, source control of bacteremia, absence of hypotension, and clinical resolution of all signs and symptoms of infection. In catheter-related bacteremia, catheters were removed on the day of diagnosis. Mortality was related to infection provided that there was microbiological, histological, or clinical evidence of active infection.

### 2.3. Microbiological Studies

Bacteremia was defined as the isolation of pathogenic bacteria in at least one bottle of blood culture (BD BACTEC^TM^ Plus Aerobic/F and Plus Anaerobic/F), analyzed with BD BACTEC (Becton Dickinson, Sparks, MD, USA) or BacTALERT 3D (bioMérieux Inc., Durham, NC, USA), depending on the method available at each center, for a minimum incubation period of five days. MDR-GNB was defined as GNB resistant to three or more of the following categories of antibiotics: carbapenems, piperacillin/tazobactam, third and fourth-generation cephalosporins, aztreonam, fluoroquinolones, or aminoglycosides [30,31]. Microbiological identification was performed with manual biochemical and microbiological methods and/or MALDI-TOF (BD Bruker Microflex MALDI Biotyper, Bruker Daltonics, Bremen, Germany). Susceptibility testing was performed with disk diffusion (according to the CLSI recommendations) and/or Etest, VITEK II Compact (bioMérieux), PHOENIX 100 BD automated system (Becton Dickinson), and VITEK MS (bioMérieux). Carbapenemase production was investigated in carbapenem-resistant bacteria using the modified Hodge method, disk synergy tests with a carbapenem disk placed close to the boronic acid disk test for KPC, and the EDTA disk for identification of metallo-β-lactamases. The presence of genes coding for *bla*KPC and *bla*OXA-48 was investigated using a monoplex or multiplex polymerase chain reaction (PCR) using specific primers depending on the method available at each center. Multiplex PCR for *bla*VIM, *bla*NDM, *bla*IMP, *bla*KPC, and *bla*OXA-48 was used to investigate isolates at the National Reference Laboratory of Microbiology (ANLIS-Malbrán) [32]. In order to detect colonization with KPC-producing Enterobacterales, rectal swabs were routinely collected (once a week and in every pre-transplant evaluation) in 10 of the 11 centers included in the study, using chromogenic methods and/or PCR.

### 2.4. Statistical Analysis

The study population was characterized by descriptive statistics. For continuous variables, centrality (median) and dispersion (interquartile range (IQR)) measures were used according to the distribution of variables. Categorical variables were analyzed using absolute frequency and percentage. Groups were compared using the Kruskal Wallis test for continuous variables and the Fisher exact test or chi-square test for categorical variables. For all tests, a 95% level of statistical significance was used.

To identify the risk factors for 30-day mortality, a multiple logistic regression model was used. Variables with *p* < 0.05 in the univariate analysis were included in the multivariate model. All reported *p*-values are 2-tailed.

Cumulative survival incidence was estimated with the Kaplan–Meier method.

Analyses were performed with the SPSS (Statistics for Windows, Version 22.0. Armonk, NY, USA) software packages.

## 3. Results

A total of 779 high-risk neutropenic patients with hematological malignancies or hematopoietic stem cell transplants who presented Enterobacterales bacteremia were evaluated, and 541 were excluded because they did not meet the eligibility criteria. Thus, the study population consisted of 238 patients, 18 with KPC-producing Enterobacterales bacteremia treated with ceftazidime–avibactam in G1, 52 with KPC-producing Enterobacterales bacteremia treated with other antibiotics in G2, and 168 treated for ESBL-Enterobacterales bacteremia in G3 (Figure 1).

The baseline and epidemiological characteristics of bacteremia episodes in the three groups are described in Table 1.

One hundred and fifty-four patients (64.7%) were male; the median age was 48 years (IQR 34–60). Acute leukemia was the most frequent underlying disease (59.6%), followed by lymphoma (25.6%). Seventy-seven patients (32.3%) received hematopoietic stem cell transplants (59.7% allogeneic). Hematological malignancy was active in 72.2% of the patients (33.5% recently diagnosed); 76% had received chemotherapy one month prior to bacteremia episode, 30.6% had received high-dose corticosteroids, and 18.4% had received biological agents and/or anti-lymphocyte therapy. Sixty-four percent of the patients had previously received antibiotics, with piperacillin–tazobactam and carbapenems being the most common (32.3% and 20.1%, respectively). In more than 50% of G1 and G2 patients, therapy duration was >7 days. Recent and previous colonization by KPC-producing Enterobacterales were detected in 23.1% and 15.9%, respectively, being significantly higher in G1 and G2. Length of hospitalization prior to bacteremia had a median duration of 13 days (IQR 3–18). Neutropenia duration was 16 days (IQR 10–25), with no differences among groups.

The Enterobacterales most frequently isolated were *Klebsiella* sp. (60.9%) and *Escherichia coli* (26.4%), with the former being the microorganisms most frequently isolated in G1 and G2. Regarding resistance profile, 78.5% of the isolates were resistant to fluoroquinolones, 14.2% to aminoglycosides, and 13.8% to colistin, with this proportion being significantly higher in G1 and G2. Only one (0.4%) isolate was resistant to ceftazidime–avibactam. The microbiological characteristics and resistance profiles of bacteremia episodes are described in Figure 2 and Figure 3.

Table 2 shows the clinical characteristics, antibiotic therapy, and outcome of patients with Enterobacterales bacteremia according to each group. One hundred and sixty-nine (71%) episodes of bacteremia had a clinical source, with abdominal (26%), catheter-related bacteremia (17.2%) and severe mucositis (12.6%) being the most frequent. Hypotension at onset was present in 80 (33.6%) patients, with similar distribution among groups. Infection severity and mortality probability measured with Pitt and APACHE-II scores were 1 (IQR 0–2) and 14 (IQR 10–19); no differences were observed among the three groups. Empirical antibiotic therapy was combined in 42.8% of cases, with carbapenems (61.3%), colistin (28.5%), piperacillin–tazobactam (24.3%), and aminoglycosides (24.3%) being the antibiotics most commonly used. One hundred and eighty (75.6%) patients received appropriate empirical antibiotic therapy, being significantly lower in G1 and G2. Patients in G3 received definitive treatment, mostly monotherapy (94.7%), while G1 and G2 received combination antibiotic therapy in 55.5% and 100%, respectively. Carbapenems (86.1%), colistin (21.4%), and aminoglycosides (10.9%) were the antibiotics most commonly prescribed.

Intensive care unit requirement, septic shock development, and breakthrough bacteremia occurred in 27.7%, 27.7%, and 10.5% of the patients, respectively, being significantly higher in G2. One hundred and seventy-eight (74.7%) patients had a 7-day clinical response, which was lower in G2 (42.3%), and was reached in all but one patient in G1 (94.4%). The 7-day mortality, 30-day mortality, and 30-day infection-related mortality were 12.6%, 21.8%, and 17.2%, respectively, but were much higher in G2 (40.3%, 53.8%, and 51.9%, respectively). In contrast, only one patient in G1 (5.5%) died due to bacteremia on day 30. The differences observed in treatment and outcome among the 3 groups are highlighted in Table 2. Cumulative survival in each group is depicted in Figure 4.

The results of univariate and multivariate analyses of risk factors for 30-day mortality are shown in Table 3. In multivariate analyses, Pitt score > 4 and KPC-producing Enterobacterales bacteremias treated with other antibiotics were independent predictors of mortality, while the 7-day clinical response was a protective factor for survival.

## 4. Discussion

The study assessed the outcome of high-risk neutropenic patients with multidrug-resistant Enterobacterales bacteremia, depending on their resistance mechanism and antibiotic therapy provided. The study included patients with KPC-producing Enterobacterales bacteremia who received at least 48 h of appropriate definitive antibiotic therapy with ceftazidime–avibactam or other antibiotics. They were compared with patients administered appropriate definitive therapy for ESBL-producing Enterobacterales bacteremia. The cohort comprised patients with hematological malignancies or hematopoietic stem cell transplants, with a high proportion presenting active underlying diseases. Length of neutropenia duration, severity of clinical presentation, and type of clinical source of bacteremia were similar among groups. *Klebsiella* spp. was isolated from most KPC-producing Enterobacterales bacteremia, and they had a high level of resistance to the antibiotics most commonly used. Patients with KPC-producing bacteremias treated with other antibiotics had higher 7-day mortality, 30-day overall mortality, and infection-related mortality than those treated with ceftazidime–avibactam. The latter had a 7-day clinical response and infection-related mortality similar to those treated for ESBL-enterobacterales bacteremia.

Overall, 30-day mortality rates after carbapenem-resistant Enterobacterales bacteremia in patients with hematological malignancies and hematopoietic stem cell transplant range from 50% to 72.7% [33,34,35]. In addition, these infections were identified as an independent predictor factor for 30-day mortality (OR: 7.4; 95% CI: 3.4–15.9; *p* < 0.001) [9].

These findings highlight the importance of the early identification of patients with carbapenem-resistant Enterobacterales bacteremia and the administration of the best therapeutic option. In this regard, a clinical score has been recently developed to identify patients at risk for these infections [36]. This study could detect three risk factors for carbapenem-resistant Enterobacterales bacteremia, with the points assigned to each of them being recent colonization with KPC-producing Enterobacterales: 5 points, previous antibiotic treatment > 7 days: 2 points, and ≥10 days of hospitalization until bacteremia: 2 points. With a cut-off of 7 points, a specificity of 98.43% and a positive predictive value of 77.7% were obtained, with a good predictive performance.

In addition to the assessment of all the potential risk factors of each patient, including the local epidemiology of antibiotic resistance, this score could contribute to a more accurate approach to prescribing an appropriate empirical therapy.

Two larger studies evaluated the outcome of patients treated with ceftazidime–avibactam for carbapenemase-producing-Enterobacterales infections. In the first, which included 391 episodes of KPC-producing *K*. *peumoniae* bacteremia treated with ceftazidime–avibactam, the 30-day overall mortality was 26.3%; however, only 5.6% of the patients were neutropenic [17]. In the second, which included 339 patients, the 30-day crude mortality in subjects with bacteremia was 13.9% in 72 patients treated with ceftazidime–avibactam vs. 30.8% in 39 treated with the best available therapy (*p* = 0.03). The multivariate analysis showed that ceftazidime–avibactam treatment was a protective factor for survival (OR 0.41, 95% CI 0.20–0.80, *p* = 0.01). As in the previously mentioned study, only 5.3% of the patients were neutropenic [18]. These studies showed no differences in mortality whether the patients received ceftazidime–avibactam as monotherapy or combination therapy.

Our results are in accordance with the previous studies since patients treated with ceftazidime–avibactam presented lower overall mortality than those who received other antibiotics. In addition, although our population included high-risk neutropenic patients with severe clinical sources of bacteremia and high median APACHE II scores, they presented very low infection-related mortality. Moreover, almost 50% received definitive ceftazidime–avibactam as monotherapy.

Two retrospective studies compared ceftazidime–avibactam therapy vs. other antibiotics against carbapenemase-producing Enterobacterales bacteremia in hematologic patients. The first is a multicenter study that included 8 patients treated with ceftazidime–avibactam (5 of them neutropenic) and compared them with 23 treated with other antibiotics [20]. They found a higher clinical cure at day 14 in the ceftazidime–avibactam group (85.7% vs. 34.8%, *p* = 0.03) and lower 30-day mortality (25% vs. 52.2%), with no statistical significance, probably due to the small sample size. Our study found lower 7- and 30-day mortality in patients who received ceftazidime–avibactam.

The second is a single-center study that evaluated ceftazidime–avibactam-based therapy vs. colistin-based therapy for the empirical treatment of febrile neutropenia in patients colonized by KPC-producing *K*. *pneumoniae* [21]. This study included 94 episodes of febrile neutropenia in 55 leukemic patients. They found 22 episodes with KPC-producing *Klebsiella pneumoniae* bacteremia (11 in each group), and no difference in clinical success between patients with ceftazidime–avibactam-based therapy vs. colistin-based therapy was found (91% vs. 86%). Unlike our study, no death was observed in the ceftazidime–avibactam group; however, all patients received adequate empirical treatment. Nevertheless, in our cohort, only one patient died from infection.

Our study and all those previously described showed better outcomes with ceftazidime–avibactam therapy for KPC-producing Enterobacterales bacteremia in different populations. These findings could be associated with in vitro and in vivo activities, PK/PD (pharmacokinetic/pharmacodynamic) properties, resistance, and clinical efficacy.

The ceftazidime–avibactam combination contains a broad spectrum diazabicyclooctane β-lactamase inhibitor, avibactam, which binds reversibly to class A β-lactamases including KPC carbapenemases, class C β-lactamases, and OXA-48 carbapenemases. This antibiotic has potent in vitro activity against Enterobacterales and *Pseudomonas aeruginosa*, including isolates from cancer patients [37,38]. Ceftazidime–avibactam’s in vivo activity was demonstrated in murine (including neutropenic) models, and proved pharmacodynamically predictable according to the isolated bacteria minimum inhibitory concentration values [37,39]. Moreover, simulated patients showed adequate PK/PD target attainment with approved dosages [40]. All the properties previously mentioned could contribute to the clinical efficacy of ceftazidime–avibactam.

Conversely, several caveats exist regarding the antibiotics most commonly used for treating KPC-producing Enterobacterales. Colistin is highly nephrotoxic and has a wide variability in plasma concentration, especially in critically ill patients. There are uncertainties regarding doses, and susceptibility testing should be performed with broth micro-dilution. In addition, combined treatment is recommended given its limited clinical efficacy [41,42]. Tigecycline has bacteriostatic activity and low steady-state concentrations in serum at the currently recommended doses, which preclude its use for the treatment of bacteremia. Like colistin, tigecycline is used in combination therapy for treating carbapenem-resistant Enterobacterales [43]. Aminoglycosides have been associated with a high risk of nephrotoxicity and poor outcomes when used as monotherapy for treating GNB bacteremia [44]. Data regarding the clinical efficacy of fosfomycin are limited; in addition, the development of resistance during treatment is another issue to be considered [45]. Finally, as shown in our study, a high level of antibiotic resistance is now frequently observed worldwide [46]. The reported studies are mostly retrospectively designed and compare ceftazidime–avibactam with the abovementioned antibiotics and their limitations, which could lead to potential confounders.

We acknowledge several limitations of the present study. First, as it was an observational study, treatment was not randomly assigned, which could bias the final results. Notwithstanding that, the most important variables associated with the outcome, such as age, type and stage of underlying hematological malignancy, immunosuppressive drugs received, neutropenia duration, clinical source of bacteremia, and clinical scores correlated with the risk of dying, were included and were comparable among groups. Second, the patients with KPC-producing Enterobacterales treated with other antibiotics received different drugs, making it difficult to compare with those who were treated with ceftazidime–avibactam. However, the former group received definitive combined therapy, which was associated with better outcomes than monotherapy in previous published studies [8]. Third, the number of patients treated with ceftazidime–avibactam included in this study is relatively small. Nonetheless, this is the most extensive comparative study that evaluates the clinical benefits of this vs. other antibiotics exclusively in this population. Fourth, we were unable to demonstrate that treatment with ceftazidime–avibactam was a protective factor for survival. Nevertheless, the multivariate analysis showed that KPC-producing Enterobacterales bacteremias treated with other antibiotics was independently associated with 30-day mortality. In addition, a 7-day clinical response was a protective factor for survival, which was achieved in all but one patient treated with ceftazidime–avibactam.

The strengths of our study rely on its being the first prospective, multicenter study carried out on high-risk neutropenic patients with hematological malignancies or hematopoietic stem cell transplantation, where a large number of multidrug-resistant Enterobacterales bacteremia episodes have been included. Thus, the real impact of introducing ceftazidime–avibactam therapy to high-risk neutropenic patients with this kind of infection could be largely assessed.

Initiating ceftazidime–avibactam therapy in high-risk neutropenic patients with KPC-producing Enterobacterales bacteremia can improve survival. More importantly, as we have clinical diagnostic tools to identify patients at risk for developing these infections, they could probably benefit from ceftazidime–avibactam as an empirical treatment.

In summary, our study evidences the superiority of ceftazidime–avibactam over other antibiotics in high-risk neutropenic patients with KPC-producing Enterobacterales bacteremia, who further show outcomes similar to those treated for ESBL-Enterobacterales bacteremia. Randomized studies are needed to support these findings and confirm the clinical benefits of ceftazidime–avibactam in this population.

## Figures and Tables

**Figure 1 microorganisms-12-00195-f001:**
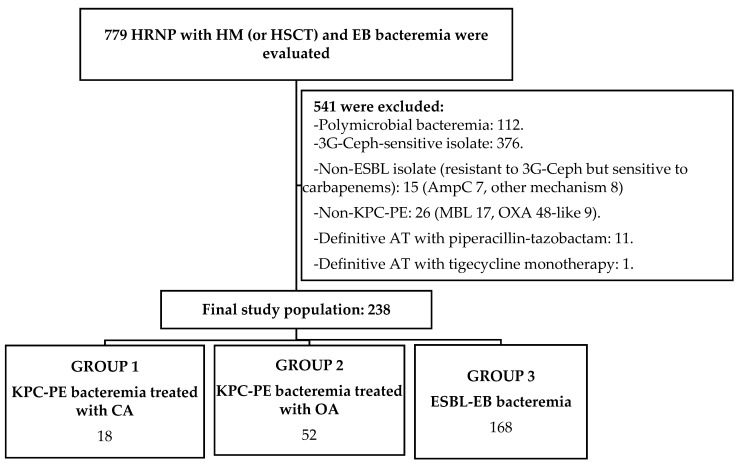
Flowchart to show the study population selection. Abbreviation: HRNP, high-risk neutropenic patients; HM, hematologic malignancies; HSCT, hematopoietic stem cell transplants; EB, Enterobacterales; 3G-Ceph, third-generation cephalosporins; ESBL, extended-spectrum beta-lactamases; KPC-PE, *Klebsiella pneumoniae* carbapenemase-producing Enterobacterales; AT, antibiotic treatment; CA, ceftazidime–avibactam; OA, other antibiotics.

**Figure 2 microorganisms-12-00195-f002:**
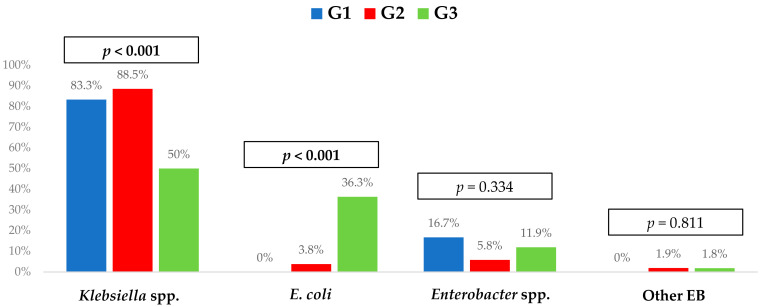
Microbiological characteristics of the 238 bacteremia episodes included, according to each bacterial and treatment group. G1, group 1 (KPC-PE bacteremia treated with ceftazidime–avibactam); G2, group 2 (KPC-PE bacteremia treated with other antibiotics); G3, group 3 (ESBL-Enterobacterales bacteremia). ESBL, extended-spectrum beta-lactamases; KPC-PE, *Klebsiella pneumoniae* carbapenemase-producing Enterobacterales; EB, Enterobacterales; other EB: *Serratia* spp., *Salmonella* spp., *Citrobacter* spp.

**Figure 3 microorganisms-12-00195-f003:**
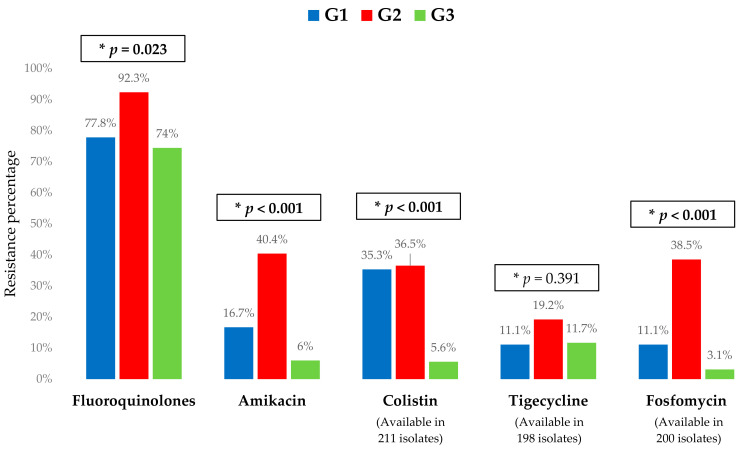
Resistance profile (% resistant to each antibiotic) of the 238 bacteremia episodes included, according to each bacterial and treatment group. G1, group 1 (KPC-PE bacteremia treated with ceftazidime–avibactam); G2, group 2 (KPC-PE bacteremia treated with other antibiotics); G3, group 3 (ESBL-Enterobacterales bacteremia). ESBL, extended-spectrum beta-lactamases; KPC-PE, *Klebsiella pneumoniae* carbapenemase-producing Enterobacterales. * *p*-values obtained by chi-square. Bold: statistically significant.

**Figure 4 microorganisms-12-00195-f004:**
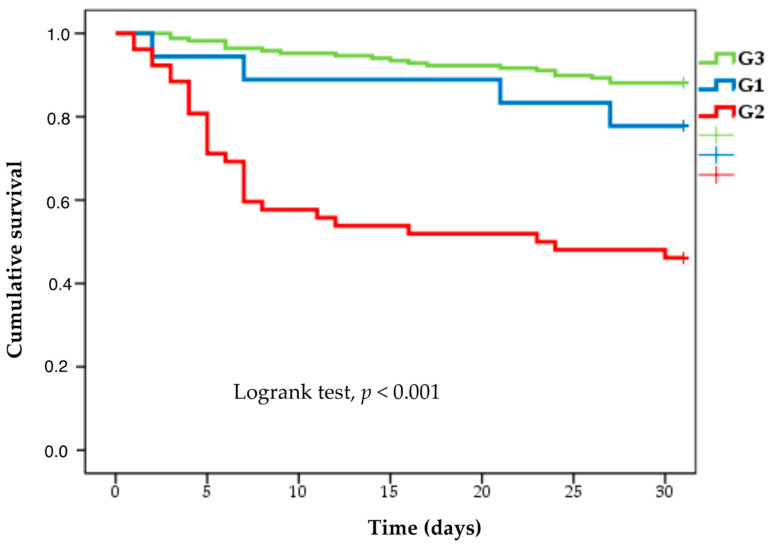
Kaplan–Meier curve showing cumulative survival according to each bacterial and treatment group. G1, group 1 (KPC-PE bacteremia treated with ceftazidime–avibactam); G2, group 2 (KPC-PE bacteremia treated with other antibiotics); G3, group 3 (ESBL-Enterobacterales bacteremia). ESBL, extended-spectrum beta-lactamases; KPC-PE, *Klebsiella pneumoniae* carbapenemase-producing Enterobacterales.

**Table 1 microorganisms-12-00195-t001:** Baseline and epidemiological characteristics.

Variables	G1 n = 18 n (%)	G2 n = 52 n (%)	G3 n = 168 n (%)	*p*-Value *
Age (years) (median, IQR)	48 (44–53)	50 (40–60)	48 (32–59)	0.56
Male gender	12 (66.7)	35 (67.3)	107 (63.7)	0.87
Charlson comorbidity index (median, IQR)	2 (2–3)	2 (2–4)	2 (2–3)	0.51
Hematological diseases				
Acute leukemia	13 (72.2)	33 (63.5)	96 (57.1)	0.38
Lymphoma	4 (22.2)	12 (23.1)	45 (26.8)	0.82
Multiple myeloma	1 (5.6)	2 (3.8)	11 (6.5)	0.77
Myelodysplastic syndrome	0 (0)	5 (9.6)	10 (6)	0.33
Other diseases	0 (0)	0 (0)	6 (3.6)	0.27
HSCT	5 (27.8)	16 (30.8)	56 (33.3)	0.86
Allogeneic HSCT	4 (22.2)	10 (19.2)	32 (19)	0.95
Stage of underlying cancer				
Recently diagnosed	7 (38.9)	28 (53.8)	45 (26.8)	**0.001**
Complete remission	4 (22.2)	12 (23.1)	50 (29.8)	0.55
Partial remission	0 (0)	3 (5.8)	22 (13.1)	0.10
Refractory	3 (16.7)	2 (3.8)	14 (8.3)	0.21
Relapse	4 (22.2)	7 (13.5)	37 (22)	0.39
Treatment of the underlying disease				
Chemotherapy (1 month prior to bacteremia)	14 (77.8)	41 (78.8)	126 (75)	0.84
Radiotherapy (1 month prior to bacteremia)	2 (11.1)	2 (3.8)	11 (6.5)	0.49
High-dose corticosteroids	2 (16.7)	24 (46.2)	46 (27.4)	**0.01**
Anti-lymphocyte drugs	6 (33.3)	9 (17.3)	29 (17.3)	0.24
Recent hospitalization (1 month prior to bacteremia)	13 (72.2)	27 (51.9)	92 (54.8)	0.31
Neutropenia duration (days) (median, IQR)	15 (10–31)	17 (14–23)	14 (10–25)	0.46
Previous antibiotic use	14 (77.8)	38 (73.1)	100 (59.5)	0.09
Previous piperacillin–tazobactam use	5 (27.8)	16 (30.8)	56 (33.3)	0.85
Previous carbapenem use	8 (44.4)	18 (34.6)	22 (13.1)	**<0.001**
Previous antibiotic use > 7 days	10 (55.6)	29 (55.8)	59 (35.1)	**0.01**
Fluoroquinolone prophylaxis	5 (27.8)	12 (23.1)	31 (18.5)	0.54
Previous colonization by KPC-PE	13 (72.2)	14 (26.9)	11 (6.5)	**<0.001**
Recent colonization by KPC-PE	18 (88.9)	28 (53.8)	11 (6.5)	**<0.001**
Previous infection by KPC-PE	7 (38.9)	9 (17.3)	23 (13.7)	**0.02**
Duration of hospitalization until bacteremia (days) (median, IQR)	27 (21–39)	32 (21–47)	30 (21–45)	0.57

Abbreviation: G1, group 1 (KPC-PE bacteremia treated with ceftazidime–avibactam); G2, group 2 (KPC-PE bacteremia treated with other antibiotics); G3, group 3 (ESBL-Enterobacterales bacteremia); IQR, interquartile range; HSCT, hematopoietic stem cell transplant; ESBL, extended-spectrum beta-lactamases; KPC-PE, *Klebsiella pneumoniae* carbapenemase-producing Enterobacterales. * *p*-values obtained using chi-square for categorical variables and Kruskal Wallis test for continuous variables. Bold: statistically significant.

**Table 2 microorganisms-12-00195-t002:** Clinical characteristics, antibiotic therapy, and outcome.

Variables	G1 n = 18 n (%)	G2 n = 52 n (%)	G3 n = 168 n (%)	*p*-Value *
Nosocomial bacteremia	17 (94.4)	51 (98.1)	133 (79.1)	**0.002**
Healthcare-associated bacteremia	1 (5.6)	1 (1.9)	28 (16.7)	**0.01**
Community-acquired infection	0 (0)	0 (0)	7 (4.2)	0.22
Bacteremia with clinical source	10 (55.6)	32 (61.5)	127 (75.6)	**0.04**
Abdominal infection	5 (27.8)	14 (26.9)	43 (25.6)	0.96
Central venous catheter infection	4 (22.2)	8 (15.4)	29 (17.3)	0.80
Respiratory infection	2 (11.1)	2 (3.8)	14 (8.3)	0.47
Severe mucositis	0 (0)	8 (15.4)	22 (13.1)	0.22
Perianal infection	1 (5.6)	1 (1.9)	6 (3.6)	0.73
Urinary tract infection	0 (0)	1 (1.9)	7 (4.2)	0.52
APACHE II score on the day of bacteremia (median, IQR)	15 (13–19)	13 (8–18)	15 (11–19)	0.14
Pitt score on the day of bacteremia (median, IQR)	1 (0–2)	1 (0–3)	0 (0–2)	0.37
Empirical Antibiotic Therapy				
Cefepime	0 (0)	0 (0)	21 (12.5)	**0.008**
Piperacillin–tazobactam	3 (16.7)	6 (11.5)	49 (29.2)	**0.02**
Carbapenem	7 (38.9)	44 (84.6)	95 (56.5)	**<0.001**
Amikacin	4 (22.2)	11 (21.2)	43 (25.6)	0.78
Colistin	6 (33.3)	30 (57.7)	32 (19)	**<0.001**
Tigecycline	3 (16.7)	16 (30.8)	0 (0)	**<0.001**
Ceftazidime–avibactam	9 (50)	1 (1.9)	4 (2.4)	**<0.001**
Appropriate EAT	12 (66.7)	24 (46.2)	144 (85.7)	**<0.001**
Combined EAT	10 (55.6)	34 (65.4)	58 (34.5)	**<0.001**
Definitive Antibiotic Therapy				
Ceftazidime–avibactam	18 (100)	0 (0)	0 (0)	**<0.001**
Carbapenem	0 (0)	39 (75)	166 (98.8)	**<0.001**
Amikacin	2 (11.1)	21 (40.4)	3 (1.8)	**<0.001**
Colistin	8 (44.4)	36 (69.2)	7 (4.2)	**<0.001**
Tigecycline	2 (11.1)	7 (13.5)	0 (0)	**<0.001**
Fosfomycin	0 (0)	14 (26.9)	0 (0)	**<0.001**
Fluoroquinolones	0 (0)	2 (3.8)	6 (3.6)	0.71
Combined DAT	10 (55.6)	52 (100)	9 (5.4)	**<0.001**
Duration of DAT	11 (8–14)	12 (9–15)	13 (6–16)	0.61
Intensive care unit admission required	4 (22.2)	27 (51.9)	35 (20.8)	**<0.001**
Septic shock development	3 (16.7)	26 (50)	37 (22)	**<0.001**
Breakthrough bacteremia	2 (11.1)	13 (25)	10 (6)	**<0.001**
7-day clinical response	17 (94.4)	22 (42.3)	139 (82.7)	**<0.001**
7-day mortality	2 (11.1)	21 (40.4)	7 (4.2)	**<0.001**
30-day mortality	4 (22.2)	28 (53.8)	20 (11.9)	**<0.001**
Infection-related 30-day mortality	1 (5.6)	27 (51.9)	13 (7.7)	**<0.001**

Abbreviation: G1, group 1 (KPC-PE bacteremia treated with ceftazidime–avibactam); G2, group 2 (KPC-PE bacteremia treated with other antibiotics); G3, group 3 (ESBL-Enterobacterales bacteremia); ESBL, extended-spectrum beta-lactamases; KPC-PE, *Klebsiella pneumoniae* carbapenemase-producing Enterobacterales; IQR, interquartile range; EAT, empirical antibiotic therapy; DAT, definitive antibiotic therapy. * *p*-values obtained by chi-square for categorical variables and Kruskal Wallis test for continuous variables. Bold: statistically significant.

**Table 3 microorganisms-12-00195-t003:** Risk factors for 30-day mortality.

Variable	Univariate Analysis			Multivariate Analysis		
	Non-Adjusted OR	95% CI	*p*-Value	Adjusted OR	95% CI	*p*-Value
Breakthrough bacteremia	3.29	1.39–7.79	**0.007**	0.88	0.21–3.57	0.852
Inappropriate EAT	2.17	1.11–2.17	**0.023**	1.03	0.31–3.41	0.957
Bacteremia due to *Klebsiella* sp.	2.94	1.42–6.07	**0.004**	1.12	0.33–3.79	0.858
7-day clinical response	0.03	0.01–0.05	**<0.0001**	0.02	0.01–0.08	**<0.001**
PITT Score > 2	5.44	2.60–11.35	**<0.0001**	3.63	1.18–11.14	**0.025**
KPC-PE bacteremia treated with OA	7.87	3.93–15.75	**<0.0001**	8.85	2.58–30.33	**0.001**

Multiple logistic regression model. Cox and Snell R^2^ = 0.401, Nagelkerke R^2^ = 0.667. Bold: statistically significant. Abbreviation: OR, odds ratio; 95% CI, 95% confidential interval; EAT, empirical antibiotic therapy; KPC-PE, *Klebsiella pneumoniae* carbapenemase-producing Enterobacterales; OA, other antibiotics.

## Data Availability

The data presented in this study are available on request from the corresponding author.

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
