# Peer review of "Ceftazidime–Avibactam Improves Outcomes in High-Risk Neutropenic Patients with Klebsiella pneumoniae Carbapenemase-Producing Enterobacterales Bacteremia"

_microorganisms, 2024, doi:10.3390/microorganisms12010195_

Round 1
Reviewer 1 Report
Comments and Suggestions for Authors
The manuscript presented by Herrera et al. concerns a very important problem about enterobacterales bacteremia in neutropenic patients.
In general, the manuscript is a very good, valuable work and an interesting source for the scientific community.
However, it needs some polishing to be improved.
1. I would recommend adding some more new references, such as:
https://doi.org/10.1007/s40265-018-0902-x
https://doi.org/10.1093/ofid/ofz360.1924
https://doi.org/10.1128/aac.02106-16
https://doi.org/10.5144/0256-4947.2023.373
2. Improve the quality of Figure 4.
3. What are the maximum plasma concentrations for both drugs near the end of the infusion?
4. In Table 2, Do you have data about trimethoprim-sulfamethoxazole?
Reviewer 2 Report
Comments and Suggestions for Authors
The manuscript addressed the topic of ceftazidime-avibactam improving outcomes in high-risk neutropenic patients with KPC-producing Enterobacterales bacteremia. The authors did a good effort in this manuscript, however, the following points should be resolved before acceptance.
The title should be revised to accurately reflect the contents of the manuscript. The authors can think about considering other titles such as "Ceftazidime-avibactam for Improved Survival in KPC-E Bacteremia in High-Risk Neutropenic Patients"
or
"Ceftazidime-avibactam Improves Outcomes in High-Risk Neutropenic Patients with KPC-producing Enterobacterales Bacteremia"
Abstract:
- More details on the "other antibiotics" group in KPC-CPE could be helpful. Knowing the specific antibiotics used and their resistance profiles would provide a clearer picture of potential confounding factors.
- More stress should be given at the end of the abstract, about the impact/recommendation of this study
Introduction:
The introduction need to be more focused and organized for better flow.
- Start with KPC prevalence and impact on high-risk neutropenic patients.
- Discuss antibiotic resistance trends, including the rise of MBLs during the pandemic.
- Highlight the challenge of empiric therapy, emphasizing the lack of specific recommendations for KPC-CPE.
- Briefly introduce CA as a promising treatment option and mention existing studies with limitations.
- Stress on the novelty of the study.
- End with the clear study aim and rationale for comparing the three groups.
Results:
The resolution of Figures 1-4 should be improved. More details should be added to the legends to be self-explannatory.
The title of figure (1) should be under the figure itself.
G1, G2 and G3 should be identified in Figure (2) and Figure (4).
The typing of the bacterial names in Figure (2) is faint.
- Discussion
- The discussion lacks in-depth analysis of potential reasons for better outcomes with CA:
- It is suggested to address the following points:
- Consider discussing specific properties of CA (e.g., broad spectrum, beta-lactamase stability) and their contribution to efficacy.
- Highlight potential confounders in non-CA group (e.g., specific antibiotics used, resistance profiles).
- Include potential implications of the findings for clinical practice:
- Briefly discuss how these results might inform treatment decisions for KPC-CPE in high-risk neutropenic patients.
- Highlight the expected further research.
- References
- All the references should be revised to make sure all the scientific names are written italic. For example Klebsiella pneumoniae in REFERENCES 15, 16 , 30, 31 and 32
Comments on the Quality of English Language
Moderate editing of English language required
Reviewer 3 Report
Comments and Suggestions for Authors
Provide the primer sequences for blaKPC, blaOXA-48, blaVIM, blaNDM, blaIMP.
Provide the details in the methods section for how Klebsiella, E. coli, and Enterobacter were identified. For example, biochemical tests or 16S rRNA sequencing.
The authors are required to explain figure 1 in the results section.
Expand KPC (Klebsiella pneumoniae carbapenemase)
Please replace "Klebsiella pneumoniae" with "K. pneumoniae" throughout the manuscript, except for the first instance of its mention. Additionally, kindly verify if there are any references to "Escherichia coli" or other bacteria within the manuscript.
Page number 12- expand HEMABIS and SEIFEM
Use the standard format for figure 2 and 3. For example, remove the horizontal lines.
Comments on the Quality of English Language
Moderate editing of English language required
Round 2
Reviewer 2 Report
Comments and Suggestions for Authors
The manuscript is significantly improved. The authors have made all the requested changes to the satisfaction of the reviewer.